# Decoding the genetic and chemical basis of sexual attractiveness in parasitic wasps

**Weizhao Sun, Michelle Ina Lange, Jürgen Gadau, Jan Buellesbach***

Institute for Evolution & Biodiversity, University of Münster, Hüfferstr, Münster, Germany

**Abstract** Attracting and securing potential mating partners is of fundamental importance for reproduction. Therefore, signaling sexual attractiveness is expected to be tightly coordinated in communication systems synchronizing senders and receivers. Chemical signaling has permeated through all taxa of life as the earliest and most widespread form of communication and is particularly prevalent in insects. However, it has been notoriously difficult to decipher how exactly information related to sexual signaling is encoded in complex chemical profiles. Similarly, our knowledge of the genetic basis of sexual signaling is very limited and usually restricted to a few case studies with comparably simple pheromonal communication mechanisms. The present study jointly addresses these two knowledge gaps by characterizing two fatty acid synthase genes that most likely evolved by tandem gene duplication and that simultaneously impact sexual attractiveness and complex chemical surface profiles in parasitic wasps. Gene knockdown in female wasps dramatically reduces their sexual attractiveness coinciding with a drastic decrease in male courtship and copulation behavior. Concordantly, we found a striking shift of methyl-branching patterns in the female surface pheromonal compounds, which we subsequently demonstrate to be the main cause for the greatly reduced male mating response. Intriguingly, this suggests a potential coding mechanism for sexual attractiveness mediated by specific methyl-branching patterns in complex cuticular hydrocarbon (CHC) profiles. So far, the genetic underpinnings of methyl-branched CHCs are not well understood despite their high potential for encoding information. Our study sheds light on how biologically relevant information can be encoded in complex chemical profiles and on the genetic basis of sexual attractiveness.

*For correspondence:
buellesb@uni-muenster.de

**Competing interest:** The authors declare that no competing interests exist.

## Editor's evaluation

This important study reveals the genetic regulation of changes in cuticular hydrocarbon profiles in a Hymenopteran insect and links these changes with courtship behaviour and sexual attractiveness. It provides convincing empirical evidence, spanning genetic, chemical, and behavioural data. It adds valuable new perspectives on the mechanisms that underlie chemical recognition and communication systems in nature.

## Introduction

Tightly coordinated chemical signaling has repeatedly been shown to be of fundamental importance for successful reproduction in a wide range of animal species (*Blomquist and Ginzel, 2021*; *Wyatt, 2014*). Particularly insects have exploited this type of signaling as their primary mode of communication (*Buellesbach et al., 2018a*; *Greenfield, 2002*). Nevertheless, exactly how specific information such as mating status or attractiveness is encoded in the myriad of signaling molecules documented to be involved in sexual communication remains poorly understood (*Stökl and Steiger, 2017*; *Allison and Carde, 2016*). Cuticular hydrocarbons (CHCs), major components on the epicuticle of insects,

**eLife digest** Attracting a mate is critical in all species that sexually reproduce. Most animals, particularly insects, do this using chemical compounds called pheromones which can be sensed by potential mates. But how these vast range of different compounds encode and convey the information needed to secure a partner is not fully understood, and the genes that drive this complex communication mechanism are largely unknown.

To address this knowledge gap, Sun et al. studied the parasitic wasp *Nasonia vitripennis*. Like other insects, female *N. vitripennis* contain a wide range of chemical compounds on their cuticle, the outer waxy layer coating their surface. Sun et al. set out to find exactly which of these compounds, known as cuticular hydrocarbons, are involved in sexual communication.

They did this by simultaneously inactivating two related genes that they hypothesized to be responsible for synthesizing and maintaining chemical compounds on the cuticle of insects. The genetic modification altered the pattern of chemicals on the surface of the female wasps by specifically up- and down-regulating compounds with similar branching structures. The mutant females were also much less sexually attractive to male wasps.

These findings suggest that the chemical pattern identified by Sun et al. is responsible for communicating and maintaining sexual attractiveness in *N. vitripennis* female wasps. This is a significant stepping stone towards unravelling how sexual attractiveness can be encoded in complex mixtures of pheromones.

The results also have important implications for agriculture, as this parasitic wasp species is routinely used to exterminate particular fly populations that cause agricultural damage. The work by Sun et al. provides new insights into how these wasps sexually communicate, which may help scientists improve their rearing conditions and sustain them over multiple generations. This could contribute to a wider application of this more sustainable, eco-friendly alternative to destructive agricultural pesticides.

are capable of chemically encoding and conveying a wide variety of biologically relevant information (*Blomquist and Ginzel, 2021*; *Blomquist and Bagnères, 2010*). Most prominently, CHCs have been shown to play pivotal roles in sexual communication as the main cues to attract and elicit courtship from conspecific mates (*Würf et al., 2020*; *Finck et al., 2016b*) to enable discrimination of con- from heterospecific mating partners (*Shahandeh et al., 2018*; *Xue et al., 2018*) and to signal receptivity and mating status (*Berson and Simmons, 2019*; *Simmons, 2015*).

Despite such diversified CHC-encoded signals and mediated behaviors, our knowledge on exactly how CHCs encode biologically relevant information has remained surprisingly scarce. This is particularly problematic in studies considering CHC profiles in their entirety as the main signaling entities (*Berson and Simmons, 2019*; *Buellesbach et al., 2018b*). The exact compounds or their combinations actually encoding the relevant information within CHC profiles remain largely elusive, except for a few case studies mainly involving the dipteran model organism *Drosophila melanogaster*, where single unsaturated CHC compounds appear to be the main mediators in sexual communication (*Marcillac and Ferveur, 2004*; *Grillet et al., 2006*). In most other cases, chemical information appears to be encoded in a much more complex manner, involving several CHC compounds in different quantitative combinations, with no deeper understanding on the actual coding patterns conveying specific information (*Würf et al., 2020*; *Buellesbach et al., 2013*).

In addition to our limited understanding on how CHC profiles encode information, our knowledge of the genetic basis of CHC biosynthesis and its impact on sexual signaling has remained comparably restricted and biased toward the *Drosophila* model system as well (*Blomquist and Bagnères, 2010*; *Holze et al., 2021*). In short, the CHC biosynthetic pathway consists of the elongation of fatty-acyl-Coenzyme A units to produce very long-chain fatty acids that are subsequently converted to CHCs (*Blomquist and Bagnères, 2010*; *Nelson and Blomquist, 1995*). An important early switch in CHC biosynthesis is either the incorporation of malonyl-CoA or methyl-malonyl-CoA, eventually leading to the production of straight-chain (*n-*) or methyl-branched (MB-) alkanes, respectively (*Holze et al., 2021*; *Nelson and Blomquist, 1995*, *Figure 1*). It has been hypothesized that these processes are mediated by two types of fatty-acyl-synthases (FAS), microsomal for methylmalonyl-CoA and cytosolic for malonyl-CoA (*Gu et al., 1997*; *Juárez et al., 1992*). To produce olefins (mono- and

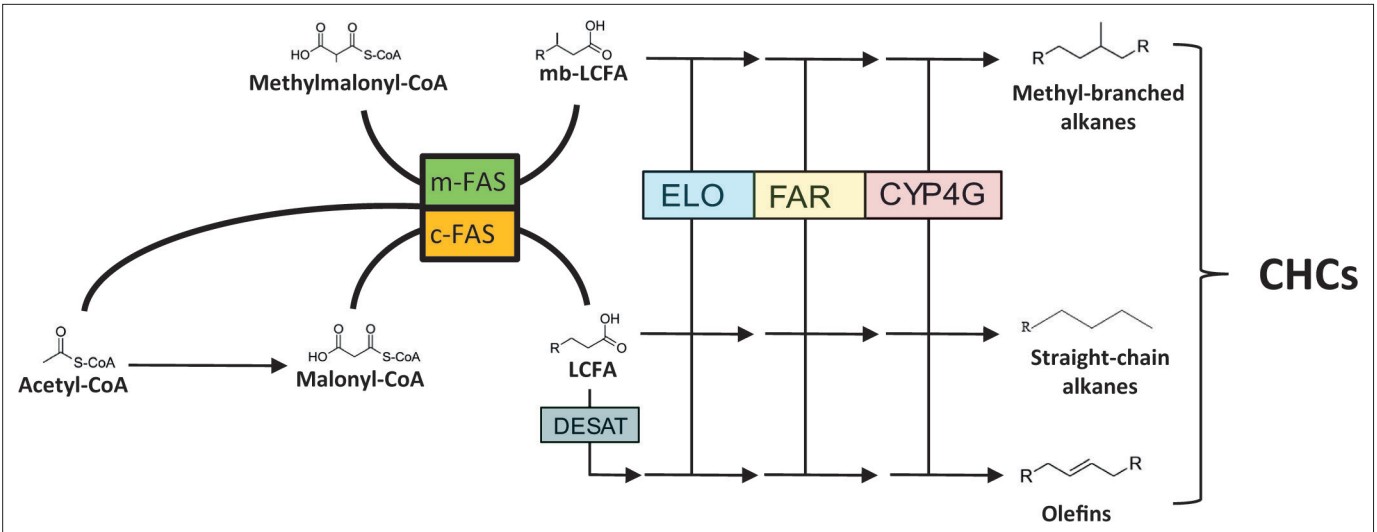

**Figure 1.** Simplified overview of cuticular hydrocarbon (CHC) biosynthesis emphasizing fatty acid synthase (FAS) catalyzed reactions. Initially, acetyl-Coenzyme A (CoA) is converted into malonyl-CoA by the enzyme Acetyl-CoA carboxylase (ACC). Then, further malonyl-CoA subunits are successively incorporated onto the acetyl-CoA primer to form long-chain fatty acids (LCFAs) catalyzed by fatty acid synthase enzymes hypothesized to be subcellularly located in the cytosol (c-FAS). For the synthesis of methyl-branched (MB)-CHCs with internal methyl groups, methyl-malonyl-CoA units are incorporated at specific chain locations instead of malonyl-CoA units, catalyzed by m-FAS enzymes whose subcellular location has been hypothesized to be microsomal. The MB or straight-chain LCFAs are then further processed through a series of biosynthetic conversions catalyzed by the elongase (ELO) enzyme complex, fatty acyl-CoA reductase (FAR) enzymes and cytochrome P450 decarboxylase (CYP4G) enzymes to either MB or straight-chain, saturated CHCs, respectively. For the biosynthesis of unsaturated CHCs (olefins), desaturase (DESAT) enzymes introduce double bonds into the fatty acyl-CoA chain between elongation steps. For a detailed description of CHC biosynthesis and the involved enzymatic reactions see *Blomquist and Ginzel, 2021* as well as *Holze et al., 2021*.

poly-unsaturated CHCs), desaturases introduce double bonds into straight-chain CHC precursors (*Blomquist and Bagnères, 2010*; *Holze et al., 2021*). In *Drosophila*, a couple of genes have been identified that mainly affect the biosynthesis and ratios of unsaturated CHC compounds that function in sexual signaling. For instance, two desaturases (*Desat1* and *DesatF*) and one elongase (*eloF*) are involved in female diene production and consequently in their functionality as main sex pheromonal compounds (*Chertemps et al., 2007*; *Chertemps et al., 2006*). Furthermore, in the Australian congeneric species *D. serrata*, the male-specific *D. melanogaster* orthologue *FASN2* has been shown to affect the biosynthesis of three MB-CHCs, among them the additional female mating stimulant 2-Me-C26 (*Wicker-Thomas et al., 2015*; *Chung et al., 2014*). Apart from these case studies limited to *Drosophila*, we know very little about the genetic basis linking CHC biosynthesis and sexual signaling in other insects (*Holze et al., 2021*).

The parasitoid jewel wasp *Nasonia vitripennis* (Hymenoptera: Pteromalidae) has emerged as a suitable model organism to combine studies on functional genetics as well as chemical communication systems in Hymenoptera (*Buellesbach et al., 2022*; *Niehuis et al., 2013*). Female CHCs serve as sexual cues capable of eliciting male courtship and copulation behavior in *N. vitripennis* (*Buellesbach et al., 2018b*; *Steiner et al., 2006*). The CHC profiles of *N. vitripennis* females exhibit a high complexity, consisting of a mixture of *n*-alkanes, *n*-alkenes, and MB-alkanes in various quantities. Specifically, the latter fraction, which makes up more than 85% of the whole profile, displays a rich diversity in methyl-branch numbers, chain lengths, and respective relative abundances, hinting at a considerable potential for encoding differential information (*Buellesbach et al., 2013*; *Steiner et al., 2006*). However, the entire female CHC profile has long been regarded as encoding their sexual attractiveness, with no single compound, compound classes, or particular patterns being identifiable as the main conveyers of the sexual signaling function (*Steiner et al., 2006*; *Buellesbach, 2018*). Moreover, despite recent advances in unraveling the genetic architecture of CHC biosynthesis and variation in the *Nasonia* genus (*Buellesbach et al., 2022*), the effects of individual genes on CHC profiles and, more importantly, on the encoded sexual signaling function, could not be determined as of yet.

In this study, we characterize the phenotypic effects of two fatty acid synthase genes whose knockdown impacts the variation of several structurally related CHC components concordant with female sexual attractiveness. CHC profiles of knockdown females primarily showed significant up- and down-regulations of MB-alkane compounds with correlated branching patterns. At the same time, these knockdown females elicited significantly less courtship and copulation attempts from conspecific males. These constitute the first hymenopteran genes with a demonstrated function in governing the variation of primarily MB-alkanes as well as communication, hinting at a chemical coding pattern for sexual attractiveness mostly conveyed by this CHC compound class.

## Results

### Knockdown of *fas5* significantly reduced the gene expression of *fas5* and *fas6*

We conducted gene expression analysis in adult wasps to assess the impact of dsRNAi micro-injection specifically targeting the *N. vitripennis* fatty acid synthase gene *fas5*. The results revealed a significant reduction in the relative expression of our target gene *fas5* compared to the control groups (*Figure 2H*), whereas the other previously published and characterized *fas* genes (*fas1–4*) remained unaffected in their expression (*Figure 2—figure supplement 1*, *Supplementary file 2*). However, the expression of *fas6*, a previously uncharacterized *fas* gene, was also significantly downregulated in our *fas5* dsRNAi knockdown individuals (*Figure 2H*). The dsRNAi off-target analysis showed that 24% of 19-mers from the *fas5* dsRNA sequence matched to the *fas6* transcript. Notably, *fas6* is localized next to *fas5* and shares high sequence similarity with the latter (90.95% at the mRNA and 88.06% at the amino acid level) (*Figure 2—figure supplement 2*). This indicates that both genes evolved by tandem gene duplication.

### Knockdown of *fas5* dramatically alters CHC profile composition

DsRNAi micro-injection into female *N. vitripennis* at the pupal stage resulted in a striking CHC profile shift in adult females, most prominently displayed in altered MB-alkane patterns (*Figure 2A, B*, *Supplementary file 3*), while overall CHC amounts and *n*-alkane quantities remained unaffected (*Figure 2—figure supplement 3A, B*). More specifically, *fas5* knockdown significantly increased the absolute quantities (ng) of MB-alkanes with their first methyl branches positioned on the 3rd (35.07 ± 9.47) and 5th (96.34 ± 33.22) C-atom, compared to both WT (wild type) (24.08 ± 5.6 and 40.06 ± 11.79) and GFP (green fluorescent protein) RNAi females (24.75 ± 7.58 and 40.19 ± 12.37), respectively (*Figure 2C, D*). Conversely, MB-alkanes with their first methyl branches on the 7th C-atom position significantly decreased in *fas5* knockdowns (5.27 ± 2.02) compared to WT (34.3 ± 11.92) and GFP RNAi females (40.15 ± 17.1) (*Figure 2E*). Other MB-alkanes with their first methyl branches mainly positioned on the 9th, 11th, 13th, and 15th also significantly decreased in *fas5* RNAi females (46.88 ± 11.67) as opposed to WT (94.87 ± 30.47) and GFP RNAi females (106.19 ± 40.39) (*Figure 2F*). Interestingly, overall MB-alkane, *n*-alkane as well as total CHC quantities remained stable in *fas5* knockdown females compared to the controls, respectively, with no significant changes (*Figure 2G*, *Figure 2—figure supplement 3A, B*). Lastly, *n*-alkene quantities, generally only occurring in negligible quantities in *N. vitripennis* females (*Buellesbach et al., 2022*), also increased significantly in *fas5* RNAi females (8.74 ± 3.52) compared to WT (2.09 ± 0.94) and GFP RNAi females (2.33 ± 0.85) (*Figure 2—figure supplement 3C*).

Male CHC profiles were also dramatically affected by *fas5* knockdowns (*Figure 2—figure supplement 4*). Whereas *n*-alkenes and MB-alkanes with their first methyl group at the 5th position are similarly upregulated in knockdown males and females (and MB-alkanes with the first methyl group at the 7th position similarly downregulated), overall CHC quantities as well as *n*-alkane quantities are significantly higher in knockdown males as opposed to knockdown females (compare *Figure 2—figure supplement 3A, B* to *Figure 2—figure supplement 4B, C*). Furthermore, CHCs with their first methyl groups at the 3rd and 9th (and higher) positions appear to be differentially affected as well: While total amounts are significantly different in knockdown females compared to both WT and GFP controls (*Figure 2C, F*), their quantity is not significantly different in knockdown males from GFP controls, but from WT controls (*Figure 2—figure supplement 4E, H*). However, as male CHCs have

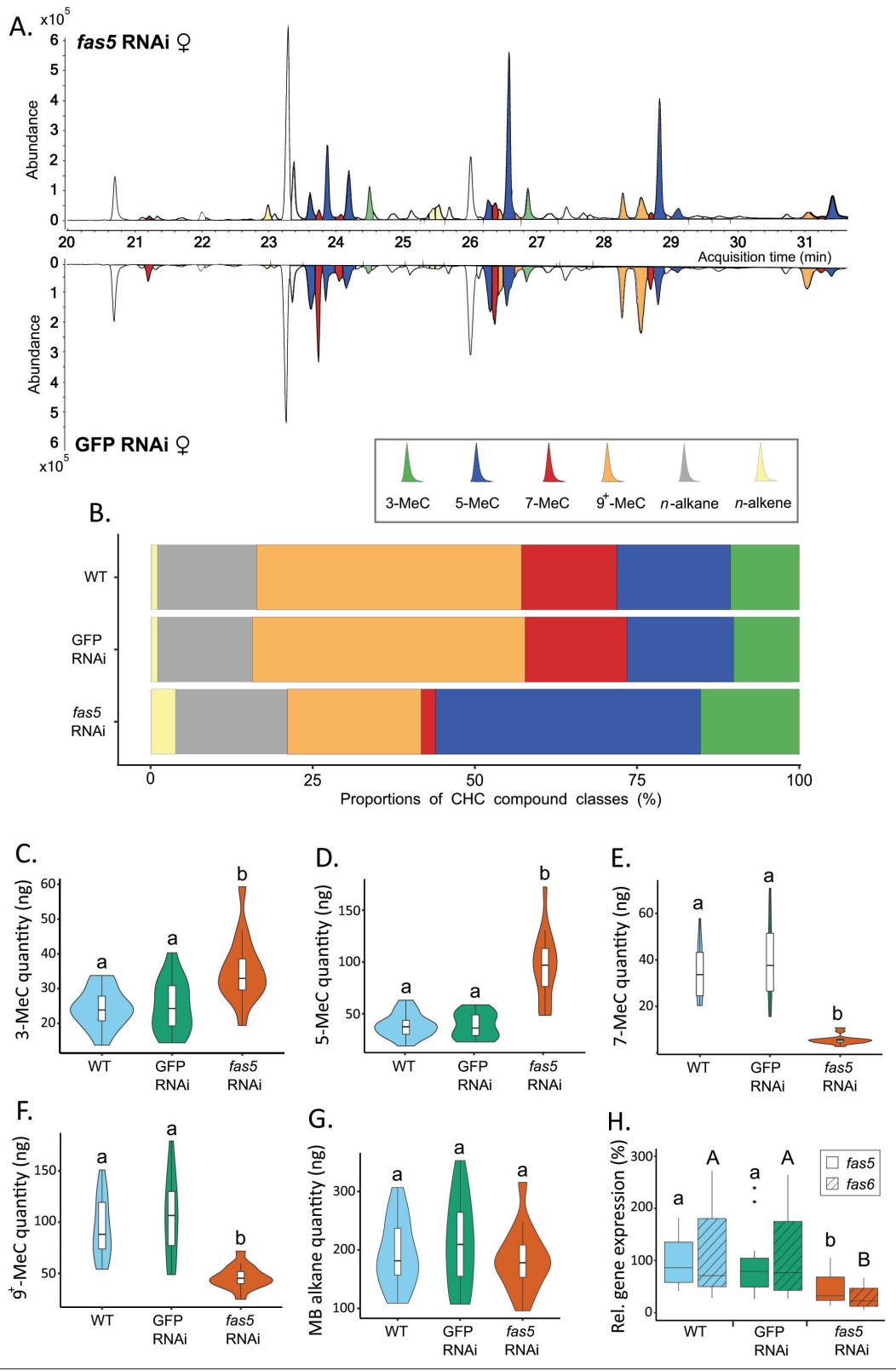

**Figure 2.** *Fas5* knockdown primarily alters quantities and ratios of methyl-branched (MB) cuticular hydrocarbons (CHCs) with specific branching patterns in females. (**A**) Chromatogram comparison of surface extracts from single female *N. vitripennis* wasps injected with *fas5* dsRNA (top) and GFP (green fluorescent protein) dsRNA (bottom). CHC compound peaks with significantly different quantities in *fas5* knockdown vs. GFP control females

*Figure 2 continued on next page*

*Figure 2 continued*

are indicated in color (compare to **Supplementary file 3**). Different colors are used for MB-alkanes with their first methyl group at positions 3-, 5-, 7-, and 9⁺ (also including positions 11-, 13-, and 15-) as well as *n*-alkanes and *n*-alkenes. (**B**) Average relative abundances (%) of different CHC compound classes (as indicated in A) compared between wild type (WT, *N* = 14), control knockdown (GFP, *N* = 15), and *fas5* knockdown (*fas5*, *N* = 15) female wasps (sample sizes per group remain consistent from here on). (**C**) Average absolute quantities (in ng) of MB-alkanes with their first methyl group at the 3rd C-atom position (3-MeC) compared between wild type (WT), control knockdown (GFP) and *fas5* knockdown (*fas5*) female wasps, indicated by blue, green, and orange violin plots, respectively (plot colors and group designations remain consistent from here on). (**D**) Average absolute quantities of MB-alkanes with their first methyl group at the 5th C-atom position (5-MeC) (**E**) Average absolute quantities of MB-alkanes with their first methyl group at the 7th C-atom position (7-MeC). (**F**) Average absolute quantities of MB-alkanes with their first methyl group at the 9th (as well as 11th, 13th, and 15th, indicated as 9⁺-MeC) C-atom position. (**G**) Average absolute quantities of total MB-alkanes. (**H**) Relative expression of *fas5* (plain) and *fas6* (hatched) in WT, GFP, and *fas5* dsRNAi females (*N* = 15 for WT and *fas5* RNAi; *N* = 16 for GFP RNAi), indicated by blue, green and orange boxplots, respectively. Significant differences ($p < 0.05$) were assessed with Benjamini–Hochberg corrected Mann–Whitney *U*-tests in (**A**) and (**C**)–(**H**) and are indicated by different letters, in (**H**) significant differences for *fas5* expression vs. controls are indicated by lower case letters, and *fas6* expression vs. controls by uppercase letters.

The online version of this article includes the following figure supplement(s) for figure 2:

**Figure supplement 1.** *Fas5* knockdown does not affect the expressions of the previously published *Nasonia vitripennis* fas genes.

**Figure supplement 2.** *Fas5* is an adjunct to *fas6* in the *Nasonia vitripennis* genome, exhibiting high sequence similarity.

**Figure supplement 3.** *Fas5* knockdown does not change total cuticular hydrocarbon (CHC) and *n*-alkane quantities but increases *n*-alkene quantities in females.

**Figure supplement 4.** *Fas5* knockdown increases total cuticular hydrocarbons (CHC) quantity and shifts the ratios of methyl-branched (MB) CHCs with specific branching patterns in males.

not been shown to function in chemical communication in *N. vitripennis*, those were not investigated further.

### *fas5* knockdown decreases attractiveness of female CHC profiles

*Fas5* RNAi and control (WT and GFP) females elicited antennation from similar proportions of males, while a significantly reduced proportion of males performed courtship and copulation toward *fas5* RNAi females (~50%) compared to controls (~90%) (**Figure 3B**). Furthermore, 70% of the males rejected *fas5* RNAi females at least once, a behavior which was not present at all toward control females (**Figure 3C**). To further minimize active female involvement in mate choice and increase male reliance on chemical cues (**Buellesbach et al., 2018b**; **Buellesbach et al., 2013**), we subsequently offered differentially treated female dummies (i.e., freeze-killed females) to WT males. Similar proportions of males showed antennation toward freeze-killed *fas5* RNAi, GFP RNAi, and WT females (**Figure 3D**). However, significantly fewer males performed courtship and copulation behavior toward freeze-killed *fas5* RNAi females (~10%, respectively) compared to both controls (>75%, respectively) (**Figure 3D**). To further test whether this dramatic reduction in sexual attractiveness is related to the altered surface profile, we proceeded to manipulate the CHC profile of female dummies. Female dummies either had their chemical profiles completely removed (i.e., cleared) or we reconstituted initially cleared WT female dummies with chemical profiles from *fas5* RNAi, GFP RNAi, or WT females, respectively. Likewise, no significant difference was found on the proportion of males that performed antennation toward female dummies of all treatments (**Figure 3E**). However, significantly less males performed courtship toward female dummies that were cleared of their chemical profiles (0%) and *fas5* RNAi-reconstituted dummies (16%), compared to WT (50%) and GFP RNAi-reconstituted (75%) female dummies (**Figure 3E**). Furthermore, copulation attempts were initiated by less than 5% of the males toward *fas5* RNAi-reconstituted dummies, which is similar to completely cleared dummies (0%). These numbers were in both cases significantly lower compared to control dummies reconstituted with WT and GFP RNAi profiles (~40%, respectively) (**Figure 3E**). This demonstrates that the dramatic reduction in female attractiveness in *fas5* knockdown females is predominantly mediated by their altered chemical profiles.

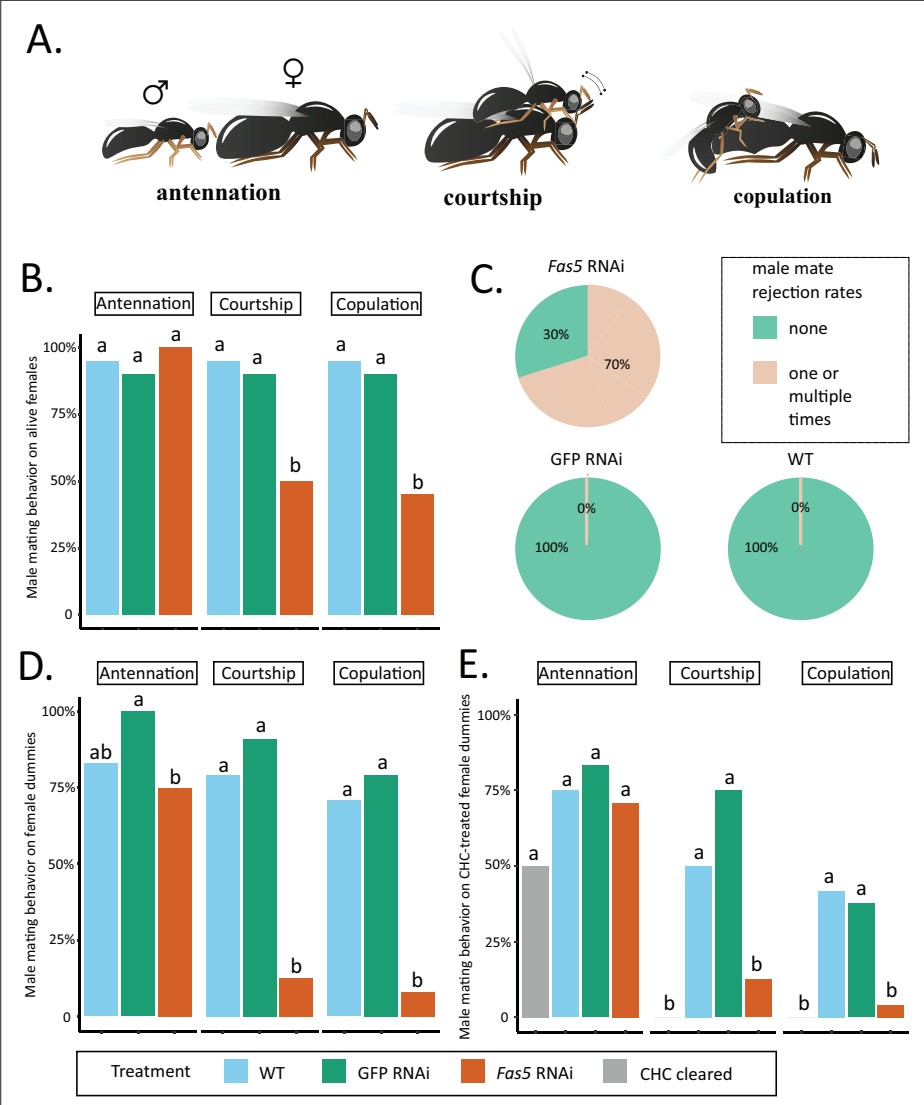

**Figure 3.** *Fas5* knockdown females elicit less courtship and copulation behaviors from WT males. (**A**) Depiction of consecutively displayed mating behavior of *N. vitripennis* males toward females, consisting of initial antennation, courtship (stereotypical head-nods on the female antennae after mounting), and actual copulation (injection of the male aedeagus into the female's genital opening) (images by Quoc Hung Le). (**B**) The proportions of males performing antennation, courtship, and copulation toward alive WT, GFP, and *fas5* RNAi females, which are marked by blue, green, and orange bar plots, N = 20 for each treatment. (**C**) Male mate rejection rates toward alive WT, GFP, and *fas5* RNAi females, separated between no (light green) and one or multiple rejections (light orange). (**D**) The proportions of males performing antennation, courtship, and copulation toward freeze-killed WT, GFP, and *fas5* RNAi females. Bar plot colors and group designations as in (B), N = 24 for each treatment. (**E**) The proportions of males performing antennation, courtship, and copulation toward either cuticular hydrocarbon (CHC) cleared (in gray) female dummies, or CHC cleared female dummies reconstituted with one female CHC profile equivalent from WT, GFP, and *fas5* RNAi females. Other treatment colors and group designations as in (B) and (D), N = 24 for each treatment. Significant differences (p < 0.05) were assessed with Benjamini–Hochberg corrected Fisher's exact tests in (B), (D) – (E) and are indicated by different letters, separated for each male mating behavior category.

## Female sexual attractiveness is mainly governed by MB-alkanes

To further pinpoint the part of the chemical profile that is responsible for encoding sexual attractiveness, we fractionated the CHC profiles of *N. vitripennis* females and focused on the MB-alkane fraction, which displayed the most conspicuous changes in *fas5* knockdown females (*Figure 2B*). The separation process reduced both the *n*-alkane and *n*-alkene proportions to less than 1% of the

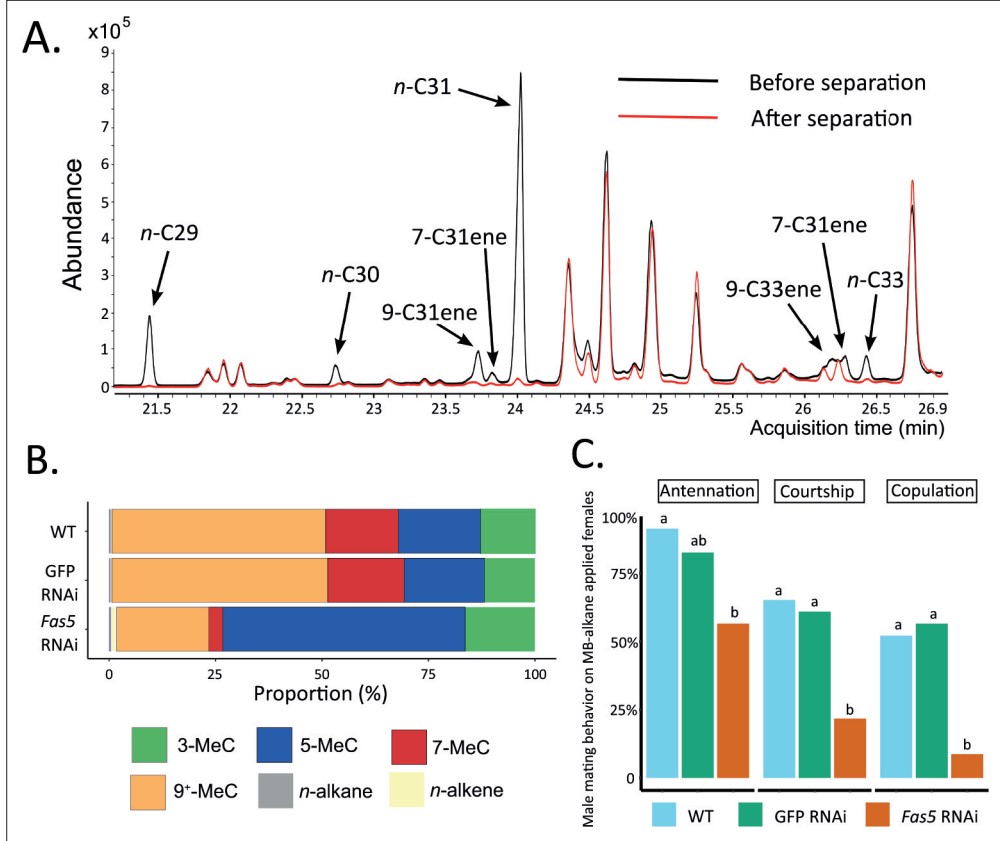

**Figure 4.** Methyl-branched (MB) alkane fraction from *fas5* RNAi females elicits less courtship and copulation from WT males. (**A**) Chromatogram comparison of representative *fas5* RNAi female cuticular hydrocarbon (CHC) profiles before (in black) and after (in red) physical separation of the MB-alkane fraction from the other compound classes (*n*-alkanes and *n*-alkenes). Individual *n*-alkane and *n*-alkene compound peaks are indicated by arrows (all other peaks correspond to MB-alkanes). Note that only the part of the *Nasonia* CHC profile where *n*-alkane and *n*-alkene compounds do occur is shown (compare to *Figure 1A*). (**B**) Average relative abundances (%) of different compound classes, including MB-alkanes with their first methyl group at positions 3-, 5-, 7-, and 9$^+$ (also including positions 11-, 13-, and 15-) as well as *n*-alkanes and *n*-alkenes, compared between wild type (WT, $N = 3$), control knockdown (GFP, $N = 3$), and *fas5* knockdown (*fas5*, $N = 3$) individuals. (**C**) The proportions of males performing antennation, courtship, and copulation toward CHC cleared female dummies reconstituted with approximately one female equivalent of MB-alkane fractions derived from WT (in blue, $N = 23$), GFP (in green, $N = 23$), and *fas5* RNAi (in orange, $N = 24$) females. Significant differences ($p < 0.05$) were assessed with Benjamini–Hochberg corrected Fisher's exact tests in (**C**) and are indicated by different letters, separated for each male mating behavior category.

The online version of this article includes the following source data for figure 4:

**Source data 1.** Methyl-branched (MB) alkane separation process dramatically decreases *n*-alkane and *n*-alkene quantities .

whole profile, allowing to focus almost exclusively on the MB-alkane fraction (*Figure 4A*; *Figure 4—source data 1*). Specifically, the MB-alkane fraction of *fas5* RNAi females maintained the dramatically higher proportions of alkanes with 3rd and 5th C-atom methyl-branch positions and lower proportions of alkanes with 7th, 9th, 11th, 13th, and 15th C-atom methyl-branch positions, compared to the respective control females' MB-alkane fractions (*Figure 4B*). We also reconstituted cleared WT female dummies with the separated MB-alkane fractions and offered them to WT males in further behavioral assays. As in our previous behavioral assays with female dummies and reconstituted whole CHC extracts, the MB-alkane fraction from *fas5* RNAi females elicited significantly less courtship and copulation attempts from WT males than the MB-alkane fractions of both WT and GFP RNAi females (*Figure 4C*). Overall, these results demonstrate that sexual attractiveness appears to be mainly encoded in the MB-alkane fraction of *N. vitripennis* females.

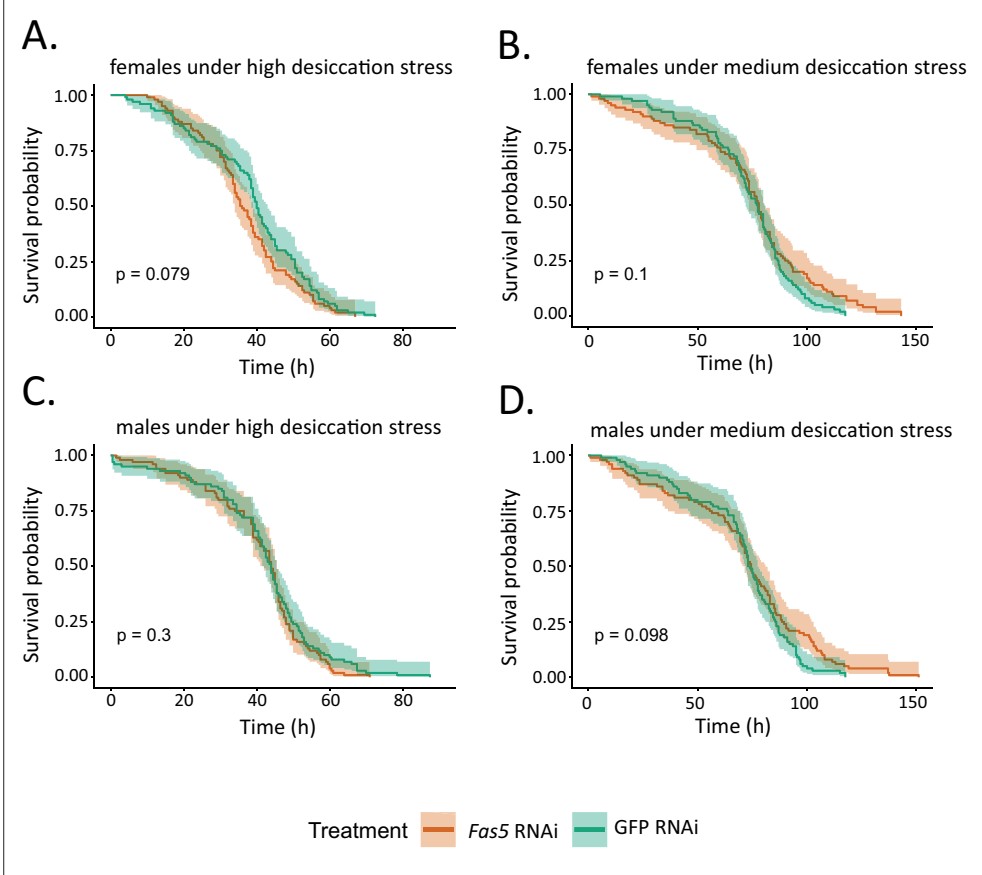

**Figure 5.** *Fas5* knockdown does not change survival times of male and female wasps under desiccation stress. (**A**) Comparison of survival probabilities along the observation time under high desiccation stress between control knockdown (GFP RNAi) and *fas5* knockdown (*fas5* RNAi) females. (**B**) Comparison of survival probabilities along the observation time under medium desiccation stress between GFP RNAi and *fas5* RNAi females. (**C**) Comparison of survival probabilities along the observation time under high desiccation stress between GFP RNAi and *fas5* RNAi males. (**D**) Comparison of survival probabilities along the observation time under medium desiccation stress between GFP RNAi and *fas5* RNAi males. *N* = 10 for each treatment. The high desiccation stress treatment was achieved with approximately 9% relative humidity, and the medium desiccation stress treatment with approximately 55% relative humidity as assessed by humidity–temperature probes. Survival probability was assessed with a Cox regression Analysis, and the colored area along the survival curve represents the 95% confident interval.

## *Fas5* knockdown does not affect desiccation resistance

Since CHC profiles also play a pivotal role in desiccation resistance (*Blomquist and Bagnères, 2010*), we further performed survival assays to explore the impact of the *fas5* knockdown on the wasps' ability to survive under different degrees of desiccation stress (*Figure 5*). Two humidity conditions were created: One with approximately 55% humidity, an ambient condition, constituting medium desiccation stress, and one with approximately 9% humidity, a highly dry condition, constituting high desiccation stress. We then tested and compared the survival times of male and female wasps treated with *fas5* RNAi and GFP RNAi (control), respectively. Unsurprisingly, males (*Figure 5B, D*) and females from both RNAi treatments survived longer under medium desiccation stress than under high desiccation stress (compare *Figure 5A, C* with *Figure 5B, D*, respectively). However, under both high and medium desiccation stress, there were no significant difference in survival probability between *fas5* and GFP RNAi treated males (*Figure 5C, D*) and females (*Figure 5A, B*).

## Discussion

In our study, we shed light on how sexual attractiveness can be encoded by differentially branched CHCs and unravel its genetic architecture to be based on two highly similar fatty acid synthase genes. Knocking down these two genes in *N. vitripennis* females leads to a consistent pattern of primarily up- and downregulated MB-alkanes with specific branching patterns. This dramatic shift is accompanied by a significant reduction of courtship and copulation behavior toward knockdown females by conspecific males which we demonstrate to be mainly determined by the altered MB-alkane fraction. This advances our understanding of how genetic information is translated into chemical information and brings us a step closer in decoding biologically relevant information from complex chemical profiles.

Most conspicuously in females, quantities of CHCs with their first methyl groups at the 7th and 9th (and higher) C-atom positions are dramatically downregulated, whereas the ones with their first methyl groups at the respective 3rd and 5th positions appear mainly upregulated, in most cases by several orders of magnitude (*Figure 2* and *Supplementary file 3*). Intriguingly, overall CHC quantities do not differ between knockdown and control individuals (*Figure 2—figure supplement 3A*), suggesting a very specific regulatory function for *fas5/fas6* in governing these opposing and potentially compensatory branching patterns. Concordantly, this emphasizes the pivotal role of both these particular genes and the wild type MB-alkane branching patterns in encoding and maintaining the attractiveness of female *N. vitripennis* CHC profiles. Moreover, there appear to be a number of sex-specific differences in the knockdown effects on the different CHC compound classes, most notably apparent in significant upregulations of both total and *n*-alkane CHC quantities in knockdown males as opposed to females and partially differentially affected MB-alkane quantities (compare *Figure 2* to *Figure 2—figure supplement 4*). Curiously, GFP dsRNAi appears to have a generally upregulating effect as opposed to WT controls despite for *n*-alkene quantities in males, which has not been reported before. However, a recent study hints at potential off-target effects of GFP dsRNAi on a small subset of *N. vitripennis* genes mainly involved in microtubule and sperm development (*Rougeot et al., 2021*). Interestingly, this appears to be also the case for the expressions of *fas1* and *fas2* gene transcripts, which are significantly upregulated in the GFP dsRNAi controls, again exclusively in males (*Figure 2—figure supplement 1B*). Though no off-target effects on CHC profile biosynthesis and variation have been reported so far for GFP dsRNAi, we cannot exclude this possibility, particularly since a trend toward higher CHC quantities in GFP dsRNAi controls as opposed to WT controls is also discernible in some cases for the females (*Figure 2E–G*). Therefore, as Rougeot et al. already hinted at , we strongly suggest alternative non-target controls in future studies on the genetics of CHC biosynthesis and variation. Concerning the partially different effects of *fas5* on CHC quantities in females and males, as no functionality in chemical communication could be attributed to the latters' CHC profiles so far, these could not be investigated any further. However, opposing sex-specific effects of CHC biosynthesis genes on the quantity of different compound classes appear to be a rather common occurrence, particularly in the insect model system *D. melanogaster* (*Holze et al., 2021*). Based mainly on research on the latter, CHC-based sexual signaling mechanisms have long been assumed to be comparatively simple, mainly mediated by two doubly unsaturated dienes in females and a mono-unsaturated *n*-alkene in males (*Marcillac and Ferveur, 2004*; *Grillet et al., 2006*). However, experimental evidence has accumulated that other CHC compounds can also complement sexual signaling in *Drosophila* in various ways (*Chung et al., 2014*; *Everaerts et al., 2010*). When regarding CHC profiles in their entirety as opposed to single compounds, direct causal links between sexual attractiveness and CHC profiles properties have rarely been experimentally demonstrated and have most often defied clear patterns (*Würf et al., 2020*; *Xue et al., 2018*). In the genus *Nasonia* and other related parasitoid wasp species, it has so far been assumed that sexual attractiveness is a trait attributed to their entire CHC profile as either present or absent depending on the studied species and also potentially reinforced by other factors such as polar cuticular compounds (*Buellesbach et al., 2013*; *Mair et al., 2017*). Our study clearly shows that CHC-mediated sexual attractiveness is primarily conveyed through a relatively complex chemical pattern with a comparatively simple genetic basis.

Interestingly, knockdown of *fas5* also upregulated *n*-alkene quantities (*Figure 2—figure supplement 3C*), which have recently been shown to have a repellent effect on *N. vitripennis* males, preventing them from engaging in homosexual courtship behavior (*Wang et al., 2022*). However, our CHC compound class separation greatly reduced the increased proportion of *n*-alkenes in *fas5* knockdown females to levels almost equivalent to those found in wild type females (0.94% and 0.91 %, respectively, in the

extract of *fas5* and WT MB fractions, compare *Figure 4—source data 1* to *Supplementary file 3*). This renders the contribution of *n*-alkenes to the sharp reduction in sexual attractiveness in *fas5* knockdown females unlikely and strongly suggests that the female sexual signaling function is mainly mediated by MB-alkanes (*Figure 4*). We argue that this CHC compound class indeed possesses the highest potential for encoding a wide variety of chemical information through the myriad of possible positions and numbers of methyl branches. In fact, a couple of studies have already hinted at MB-alkanes as the main carriers for chemical information in insect CHC profiles, providing evidence for the involvement of MB-alkanes in chemical communication processes (*Spikes et al., 2010*; *Holman et al., 2016*). This might be of particular importance in Hymenoptera, an insect order with CHC profiles largely dominated by MB-alkanes (*Kather and Martin, 2015*; *Martin and Drijfhout, 2009*) and in which both theoretical considerations and empirical evidence have accumulated for the increased complexity and high sophistication of their chemical communication systems (*Saad et al., 2018*; *Obiero et al., 2021*). Contrary to this, it has been argued that olefins (mainly *n*-alkenes and dienes) have a higher potential for encoding chemical information than MB-alkanes (*Chung and Carroll, 2015*). However, this view might have been biased and mainly informed by findings from *Drosophila*, where unsaturated compounds appear to be, in fact, the main mediators of sexual communication (*Marcillac and Ferveur, 2004*; *Ferveur, 2005*). In direct comparison, MB-alkanes constitute the dominant fraction in *Nasonia* CHC profiles (>85%) (*Buellesbach et al., 2022*) as opposed to *Drosophila* profiles (16–24%) (*Dembeck et al., 2015*). Since the split between Hymenoptera and other holometabolous insects including Diptera has been estimated to have occurred 327 mya (*Misof et al., 2014*), fundamental shifts in basic properties of both surface profile compositions and chemical signaling functionalities might be expected. Therefore, the promising role of MB-alkanes in conveying chemical information should be investigated more prominently in future studies. Through a wider evolutionary lens, it will be interesting to investigate how the present findings compare to other CHC-based communication systems in the vastly diverse insect order Hymenoptera. Notably, whereas in solitary Hymenoptera CHCs can function as contact sex pheromones, they are prominent and fundamental nestmate and caste recognition cues in eusocial Hymenoptera (*Buellesbach et al., 2018b*; *Vereecken et al., 2007*; *Leonhardt et al., 2016*). However, the main encoding mechanisms as well as the underlying genetic basis still remain largely elusive in most taxa, and our study constitutes an important stepping stone for investigating potential similarities in these important communication modalities in a larger evolutionary context. Moreover, since CHC profiles can be highly species-specific, their potential involvement in species recognition and assortative mating warrants further investigation to elucidate the exact underlying chemical and genetic differentiation mechanisms (*Buellesbach et al., 2018b*; *Wurdack et al., 2015*; *Weiss et al., 2015*).

Concerning gene orthology, *fas5* has been annotated as a homolog to *FASN3* in *D. melanogaster* (*Buellesbach et al., 2022*; *Lammers et al., 2019*). Interestingly, knockdowns of *FASN3* alone do not induce any compound changes in *D. melanogaster* CHC profiles but increase the flies' sensitivity to desiccation (*Wicker-Thomas et al., 2015*). A *FASN3* ortholog expressed in the kissing bug *Rhodinius prolixus* also contributes to desiccation resistance, but simultaneously downregulates MB- while upregulating straight-chain alkanes (*Moriconi et al., 2019*). Similarly, in the migratory locust *Locusta migratoria*, silencing two *FAS* genes decreased insect survival under desiccation stress while altering the amounts of both MB- and straight-chain alkanes (*Yang et al., 2020*). In contrast to these studies, we were not able confirm any functionality in desiccation resistance for *fas5* in both *N. vitripennis* males and females (*Figure 5*), nor did the previous studies report any impact on CHC-based chemical signaling. There are two additional *FAS* genes characterized in *D. melanogaster*: *FASN1*, which is responsible for overall CHC production with no specific effects on any particular compound classes (*Wicker-Thomas et al., 2015*) and *FASN2*, which exclusively regulates MB-alkane production in males (*Chung et al., 2014*). Although not a direct homolog, the main impact on MB-alkane variations of *FASN2* in *D. melanogaster* is comparable to that of *fas5* in *N. vitripennis*, with the exception that we were able to document effects on this compound class for both sexes (compare *Figure 2* and *Figure 2—figure supplement 4*). Such diversified functionalities of the *FAS* genes characterized so far indicate their high versatility as early mediators in the CHC biosynthesis pathway (*Figure 1*). *FAS* genes have been implied as instrumental for generating the huge diversity of different CHC profiles across insects, with high evolutionary turnover rates and differences in the specific functional recruitments of *FAS* gene family members (*Finck et al., 2016a*; *Moriconi et al., 2019*). However, *FAS* genes

are far from restricted to impact CHC biosynthesis alone and have been documented to be involved in a wide variety of other physiological processes, ranging from lipogenesis to diapause induction (*Tan et al., 2017*; *Lammers et al., 2019*; *Moriconi et al., 2019*). Therefore, specifically predicting functionalities of *FAS* genes and unambiguously associating them with CHC biosynthesis and variation has been notoriously difficult (*Holze et al., 2021*). Our study suggests a very specific affinity of *fas5* for particular methyl-branching patterns, potentially originating in an enzymatic preference for methyl-malonyl-CoA predecessors with pre-existing branching patterns (*Blomquist and Bagnères, 2010*; *Nelson and Blomquist, 1995*). To clarify this, future studies should determine whether the *fas5* gene product constitutes a soluble cytosolic or membrane-bound microsomal FAS enzyme, the latter of which has been postulated to be specific for the incorporation of methyl-malonyl-CoA subunits (*Gu et al., 1997*; *Juárez et al., 1992*, *Figure 1*).

The previously uncharacterized *fas* gene *fas6* not only shows high sequence similarity to *fas5*,but is also its physical neighbor (*Figure 2—figure supplement 2*). Moreover, both of these genes show similarly high expression patterns in wild type *N. vitripennis* wasps (*Figure 2H* and *Figure 2—figure supplement 4I*). This basically implies that these two genes originated from a tandem gene duplication event and therefore constitute paralogs of the *D. melanogaster FASN3*. Hence, we cannot state at this point whether either one of these two genes alone or both are responsible for the observed phenotypic changes. Interestingly, this mirrors a similar finding in *N. vitripennis* males concerning the biosynthesis of a lactone functioning in a long-range pheromonal blend attractive to virgin females (*Niehuis et al., 2013*). The production of this compound was unambiguously attributed to three short-chain dehydrogenase/reductase genes with high sequence similarity, which could also not be targeted independently by dsRNAi knockdowns. This demonstrates the difficulty to narrow down specific genes responsible for the biosynthesis of pheromonally active compounds when their sequences show particularly high degrees of similarity and simultaneously high expression patterns.

In conclusion, our study demonstrates the considerable impact of two highly similar fatty acid synthase genes on female sexual attractiveness in a parasitoid wasp, thereby immediately suggesting how this trait can be encoded through specific patterns of MB-alkanes. To the best of our knowledge, these are the first identified hymenopteran genes with a specific effect on MB-alkane ratios that simultaneously impact sexual attractiveness. Transcending the demonstrated impact on sexual signaling and elicited mating behavior, the present findings also substantially advance our general knowledge on the so far little investigated genetic underpinnings of MB-alkane variation and production. This particular compound class dominates the surface profiles of many insects, most notably in the ecologically and economically important insect order Hymenoptera, and harbors a considerable potential for encoding chemical information, inviting a stronger emphasis on these compounds in future studies on chemical signaling and its behavioral impact.

# Materials and methods
## *Nasonia* strain maintenance and preparation
The standard laboratory strain AsymCX of *N. vitripennis*, originally collected in Leiden, the Netherlands, was used for all experiments. The wasps were reared under 25°C, in 55% relative humidity and a light:dark cycle of 16:8 hr, leading to a life cycle of ~14 days. Pupae of *Calliphora vomitoria* (Diptera: Calliphoridae) were used as hosts.

## DsRNAi gene knockdown
For the knockdown initially designed to target *fas5*, dsRNA was synthesized following the manual of MEGA Script T7 kit (Invitrogen, Carlsbad, CA, USA), using the primer pairs listed in *Supplementary file 1*. The Quick-RNA Tissue/Insect Kit (Zymo Research, Freiburg, Germany) was used to purify the resulting dsRNA product. GFP (green fluorescent protein) dsRNA, which presumably has no known targets in the *Nasonia* genome (*Rougeot et al., 2021*), was used as a control. GFP dsRNA was synthesized from the vector pOPINEneo-3C-GFP, which was kindly donated by Ray Owens (Addgene plasmid #53534; http://n2t.net/addgene:53534; RRID: Addgene_53534). Microinjections were performed with 4–5 µg/µl (diluted in nuclease free water, Zymo Research) *fas5* and GFP dsRNA on a Femtojet micro-injector (Eppendorf, Hamburg, Germany) following the protocol published by *Lynch and Desplan, 2006*. *N. vitripennis* yellow pupae (7–8 days old after egg deposition) were gently fixed on a cover

slide using double-sided tape (Deli, Zhejiang, China), with their abdomens facing up. DsRNA mixed with 10% red food dye (V2 FOODS, Niedersachsen, Germany) was injected into the abdomens of the pupae using a thin needle, which was produced in a PC-10 puller (Narishigne Group, Tokyo, Japan) by heating a glass capillary (100 mm length × 85 µm inner diameter, Hilgenberg, Malsfeld, Germany) to 100°C, and subsequently breaking the stretched cappllary in two parts in the narrow middle section at 67°C. Individual injections were performed until the red dye has spread evenly within the abdomen of each pupa as described by *Wang et al., 2022*. The injected pupae were then stored inside a Petri dish with a piece of wet tissue at the bottom to ensure saturation with sufficient humidity for the pupae to mature and eclose. After the pupae elosed as adults, they were collected at an age of 0–24 hr and snap-frozen with liquid nitrogen, after which they were stored at −80°C for further experiments.

## DsRNAi efficiency analysis

DsRNAi knockdown efficiency was determined by quantitative PCR (qPCR), initially assessing *fas5* gene expression levels between knockdown and control individuals. RNA from each individual wasp after chemical extraction (see below) was obtained using the Quick-RNA Tissue/Insect Kit (Zymo Research, Freiburg, Germany), and reversely transcribed into complementary DNA (cDNA) utilizing the cDNA Synthesis Kit (CD BioSciences, New York, USA). As controls for the qPCR procedure, we used *N. vitripennis elongation factor 1α* (*NvEF-1α*) as a housekeeping gene, as described by *Wang et al., 2022*. The qPCR was performed in a Lightcycler480 qPCR machine (Roche, Basel, Switzerland), with a pre-incubation of 95°C for 3 min, 40 amplification cycles of 15 sec at 95°C and 60 sec of 60°C, as well as a final standard dissociation curve step to check the specificity of the amplification. The analysis of qPCR data was conducted using the ΔΔCt method (*Rao et al., 2013*). Firstly, the Ct (number of cycles required for the fluorescent signal to cross the threshold, i.e., exceeding the background level) values of the target genes were normalized to the Ct values of the housekeeping gene, yielding the respective difference between the two (ΔCt). Secondly, the ΔCt values of the knockdown treatments were normalized to the average ΔCt values of the wild type (WT) group, which resulted in ΔΔCt values. The relative expression patterns of the target genes were then determined using the formula $2^{-\Delta\Delta Ct}$. To compare the relative expressions of *fas* genes among the treatments, a sequential Mann–Whitney *U*-test was employed. Subsequently, the resulting p-values from these tests (18 in total, considering 6 *fas* genes and 3 treatments) were subjected to the Benjamini–Hochberg procedure for correction of the false discovery rate (*Benjamini and Yekutieli, 2001*).

## DsRNAi *fas5* off-target effects

The off-target effect of *fas5* dsRNA was assessed using the dsRNAi off-target prediction tool in WaspAtlas (*Davies and Tauber, 2015*), an online *N. vitripennis* genomic database. Briefly, the *fas5* dsRNA sequence was first split into all possible 19-mers, which were further matched to the *N. vitripennis* transcriptome. The transcripts to which at least one of the 19-mers matched were identified and then for each transcript the percentage of matching 19-mers was calculated.

## Chemical analysis

Chemical extractions of single wasps were performed by immersing them in 50 µl high-performance liquid chromatography (HPLC)-grade *n*-hexane (Merck, KGaA, Darmstadt, Germany) in 2-ml glass vials (Agilent Technologies, Waldbronn, Germany) on an orbital shaker (IKA KS 130 Basic, Staufen, Germany) for 10 min. Extracts were subsequently evaporated under a constant stream of gaseous carbon dioxide and then resuspended in 10 µl of a hexane solution containing 7.5 ng/µl dodecane (C12) as an internal standard. Following this, 3 µl of the resuspended extract was injected in splitless mode with an automatic liquid sampler (PAL RSI 120, CTC Analytics AG, Zwingen, Switzerland) into a gas chromatograph (GC: 7890B) simultaneously coupled to a flame ionization detector (FID: G3440B) and a tandem mass spectrometer (MS/MS: 7010B, all provided by Agilent Technologies, Waldbronn, Germany). The system was equipped with a fused silica column (DB-5MS ultra inert; 30 m × 250 µm × 0.25 µm; Agilent J&W GC columns, Santa Clara, CA, USA) with helium used as a carrier gas under a constant flow of 1.8 ml/min. The FID had a temperature of 300°C and used nitrogen with a 20-ml/min flow rate as make-up gas, and hydrogen with a 30-ml/min flow rate as fuel gas. The column was split at an auxiliary electronic pressure control (Aux EPC) module into an additional deactivated fused silica column piece (0.9 m × 150 µm) with a flow rate of 0.8 ml/min leading into the FID detector, and

another deactivated fused silica column piece (1.33 m × 150 µm) at a flow rate of 1.33 ml/min leading into the mass spectrometer. The column temperature program started at 60°C and was held for 1 min, increasing 40°C per min up to 200°C and then increasing 5°C per min to the final temperature of 320°C, held for 5 min.

CHC peak detection, integration, quantification, and identification were all carried out with Quantitative Analysis MassHunter Workstation Software (Version B.09.00/Build 9.0.647.0, Agilent Technologies, Santa Clara, CA, USA). CHCs were identified according to their retention indices, diagnostic ions, and mass spectra as provided by the total ion count chromatograms, whereas their quantifications were achieved by the simultaneously obtained FID chromatograms, allowing for the best-suited method for hydrocarbon quantification (Agilent Technologies, Waldbronn, Germany, pers. comm.) while simultaneously retaining the capability to reliably identify each compound. Absolute CHC quantities (in ng) were obtained by calibrating each compound according to a dilution series based on the closest eluting *n*-alkane from a C21-40 standard series (Merck, KGaA, Darmstadt, Germany) at 0.5, 1, 2, 5, 10, 20, and 40 ng/µl, respectively.

To compare the amount of each of the 54 identified single CHC compounds among different treatments, sequential Mann–Whitney *U*-tests were performed between all pairs of the two treatments (namely, *fas5* vs. GFP, *fas5* vs. WT and GFP vs. WT). Subsequently, the resulting 162 (54 × 3) p-values obtained from the sequential Mann–Whitney *U*-tests were Benjamini–Hochberg corrected to control for the false discovery rate (*Benjamini and Yekutieli, 2001*). Furthermore, single CHC compounds were categorized into distinct compound classes based on their chemical structures, that is, *n*-alkanes, *n*-alkenes, 3-methyl-branched alkanes (3-MeC), 5-methyl-branched alkanes (5-MeC), 7-methyl-branched alkanes (7-MeC), and 9$^+$ methyl-branched alkanes (9$^+$-Me, that also include 11-, 13-, and 15-MeC) with their first methyl branches at the indicated positions (in case of multiply branched MB-alkanes, other branching positions apart from the first were not further differentiated in this categorization). The compounds within each class were then combined to obtain a sum representing the total proportion of CHCs in each class. Sequential Mann–Whitney *U*-tests were performed to compare the compound class proportions between treatments, after which the 21 (7 × 3) p-values were adjusted again for multiple comparisons using the Benjamini–Hochberg procedure (*Benjamini and Yekutieli, 2001*).

## CHC compound class separations

Physical separation of the MB-alkane fraction from the other compound classes in *N. vitripennis* CHC profiles was performed according to an adapted protocol from *Würf et al., 2020* and *Bello et al., 2015*. CHC profiles of approximately 700 females were extracted in 9 ml HPLC-grade *n*-hexane (Merck, KGaA, Darmstadt, Germany) which was subsequently evaporated under a stream of gaseous carbon dioxide. The dried extract was re-suspended in 10 ml isooctane (99%, Sigma-Aldrich, Taufkirchen, Germany), and stirred overnight with a magnetic stirrer (Model: C-MAG HS4, IKA, Germany) after adding 2 g of activated (i.e., baked at 300°C for 2 hr) molecular sieves (5 Å, 45–60 mesh size, Merck, KGaA, Darmstadt, Germany). The molecular sieves were filtered out by loading the extract into a glass funnel (50 mm inner diameter) with glass wool (Merck, KGaA, Darmstadt, Germany) and 0.2 g silica gel (high purity grade, pore size 60 Å, 230–400 mesh particle size, Merck, KGaA, Darmstadt, Germany) containing 10% pulverized AgNO$_3$ (99.7%, Merck KGaA, Darmstadt, Germany). This procedure is devised to effectively filter out *n*-alkanes and olefins, retaining only the MB-alkane fraction in the remaining extract (*Würf et al., 2020*; *Bello et al., 2015*). The isooctane in the extract was condensed to 2 ml under a stream of gaseous carbon dioxide, from which we sampled 50 µl to estimate the quantity of the overall extract. Based on the quantity of overall MB-alkanes and single female CHC profiles, the MB-alkanes were reconstituted in hexane to a final concentration of approximately one female equivalent per 5 µl, which was saved for further behavioral assays.

## Behavioral assays

Mating behavior assays were carried out to test whether female sexual attractiveness decreased after knockdown of *fas5*. First, virgin females of three treatments (*fas5* dsRNAi, GFP dsRNAi, and WT) were offered to 0- to 48-hr-old virgin WT *N. vitripennis* (AsymCX) males, following the protocol described in *Mair et al., 2017*. A female was transferred into a transparent plastic vial (76 mm height, 10 mm diameter) that contained a male. The assay was started as soon as the male was introduced and

observed for 5 min. The males' behavior toward the females was then scored based on presence or absence of three consecutive behavioral displays: antennation (physical contact of male antennae with the female's body surface), courtship (series of stereotypic headnods and antennal sweeps after mounting the female), and actual copulation attempts, which have been established as indicators of male mate acceptance and female sexual attractiveness (*Buellesbach et al., 2018b*; *Buellesbach et al., 2013*; *Buellesbach et al., 2014*, *Figure 3A*). If the male did not initiate any further courtship or copulation behaviors after antennation, this was scored as a male mate rejection as described in *Buellesbach et al., 2014*.

To increase the focus on the male mate choice behavior in relation to female chemical cues, further behavioral assays were carried out with freeze-killed females (i.e., dummies) offered to 0- to 48-hr-old virgin WT males. In the first set of these experiments, male behavior was recorded and compared on female dummies of the three previously mentioned treatments (*fas5* dsRNAi, GFP RNAi, and WT). In the second set, female dummies were manipulated by either soaking them individually in 50 µl hexane for 3 hr, effectively removing their CHC profiles (*Buellesbach et al., 2018b*; *Buellesbach et al., 2013*) or reconstituting soaked WT female dummies with one fresh female CHC profile equivalent (resuspended in 5 µl hexane as described above for chemical analysis) from *fas5* dsRNAi, GFP dsRNAi, and WT females. In the third set, approximately one female equivalent of the separated MB-alkane fraction, prepared directly after the CHC separation process (see above), was reconstituted to soaked WT female dummies, and offered to the WT males.

All behavioral assays with female dummies were performed in a mating chamber which consisted of two identical aluminium plates (53 × 41 × 5 mm). Each plate contained 12 holes (6 mm diameter) that served as observation sites. In preparation for the recording of the behavioral assays, single female dummies were placed into each hole in one plate, while single WT males were placed into each hole in the opposite plate and immediately covered with glass slides (Diagonal GmbH & Co KG, Münster, Germany). The behavioral assays were initiated by quickly adjoining the two plates. Recordings were conducted with a Canon camera (EOS 70D, Tokyo, Japan) for 5 min in an eclosed wooden box (70 x 40 x 40 cm) with constant illumination (100Lm, LED light L0601, IKEA Dioder, Munich, Germany). Recordings of mating behaviors were further assessed and the frequencies of males' mating displays were compared among treatments with Benjamini–Hochberg corrected Fisher's exact tests (*Benjamini and Yekutieli, 2001*).

## Desiccation assays

High desiccation stress conditions were implemented by placing 0.6 g desiccant (DRIERITE, Merck, KGaA, Darmstadt, Germany) into transparent plastic vials (76 mm height, 10 mm diameter), which were air tightened with rubber plugs, resulting in low (~9%) relative humidity after 24 hr. Inside each vial, a piece of cotton and a stainless-steel grid was placed in the middle to separate the desiccant from the remaining space of the vial (~35 mm height), which served as an observation site where individual wasps were placed for the duration of the desiccation assay. Control vials with moderate desiccation stress (~55% relative humidity) were prepared similarly but without adding the desiccant. The relative humidity in the test tubes of different humidity treatments was monitored using a humidity–temperature probe (Feuchtemesssystem HYTELOG-USB, B+B Thermo-Technik GmbH, Donaueschingen, Germany) with a measurement accuracy of ±2% relative humidity at 25°C. Newly eclosed (0- to 24-hr-old) male and female wasps previously treated with either *fas5* dsRNAi or GFP dsRNAi were collected, sorted into groups of 10 per treatment and sex, respectively, and fed with honey water (Bluetenhonig, dm-drogerie markt GmbH & Co KG, Karlsruhe, Germany) for 9 hr. Afterwards, each group of wasps was randomly assigned to the previously prepared vials with either high or moderate desiccation stress. For each treatment, 10 replicates (vials) were observed. Recording was performed with a looped VLC media player (VideoLAN, Paris, France) script, initiating a 2-min recording (Logitech C920 HD PRO webcam, Logitech GmbH, München, Germany) every 2 hr, until the last wasp fell down on the grid and stopped moving. The numbers of alive wasps in each vial were assessed to build survival curves that were then compared among treatments, separately for each sex. Survival analysis was conducted using the Cox Proportional Hazards Model to evaluate the survival probability, employing the R package 'survival' (*Therneau et al., 2000*). The knockdown treatment was considered as the fixed factor, while the variation among replicates was treated as a random factor. To determine the statistical significance, the log-rank test was employed. Additionally,

the Benjamini–Hochberg procedure was applied to correct for multiple testing across the different humidity treatments (*Benjamini and Yekutieli, 2001*).

## Acknowledgements

We would like to thank Marek Golian, Stella Schummer, Anastasia Dzemiantsei, and Lisa Sewald for assisting the experiments. Besides, we appreciate Yunsheng Zhu's help in data analysis and Yidong Wang's insights in figure construction. Furthermore, we thank Sabine Nooten and Erik T Frank for helpful suggestions on the first draft of this manuscript.

## Additional information

### Funding

| Funder | Grant reference number | Author |
|---|---|---|
| Deutsche Forschungsgemeinschaft | 427879779 | Jan Buellesbach |

The funders had no role in study design, data collection, and interpretation, or the decision to submit the work for publication.

### Author contributions
Weizhao Sun, Data curation, Formal analysis, Validation, Investigation, Visualization, Methodology, Writing – original draft, Writing – review and editing; Michelle Ina Lange, Investigation; Jürgen Gadau, Resources, Supervision, Project administration, Writing – review and editing; Jan Buellesbach, Conceptualization, Resources, Formal analysis, Supervision, Funding acquisition, Validation, Visualization, Writing – original draft, Project administration, Writing – review and editing

### Author ORCIDs
Weizhao Sun (iD) http://orcid.org/0000-0003-2781-1228
Jan Buellesbach (iD) https://orcid.org/0000-0001-8493-692X

### Decision letter and Author response
Decision letter https://doi.org/10.7554/eLife.86182.sa1
Author response https://doi.org/10.7554/eLife.86182.sa2

## Additional files

### Supplementary files
• Supplementary file 1. List of primers used in the present study. All primers were designed using the Primer-BLAST tool from the National Center for Biotechnology Information (NCBI). Indicated are the respective primer names, their sequences, and their usage in the experimental protocol.

• Supplementary file 2. Gene names, gene IDs, and transcript IDs of all *fas* genes and the housekeeping gene (*elongation factor 1*a) from the present study. Note that there are two transcript IDs for *fas6* (compare to *Figure 2—figure supplement 2*).

• Supplementary file 3. Comparison of absolute quantities and relative abundances of single cuticular hydrocarbon (CHC compounds between differentially treated female wasps). Indicated are retention indices (RI), CHC compound identifications or possible configurations in case of ambiguities, their mean absolute (ng) amounts with their respective absolute standard deviations (sd) as well as their respective relative amounts (in %) compared between wild type (WT, $N = 14$), control knockdown (GFP RNAi, $N = 15$), and *fas5* knockdown (*fas5* RNAi, $N = 15$) female wasps. To compare the absolute quantities (ng) of 54 single CHC compounds among the different treatments, we employed a sequential Benjamini–Hochberg corrected Mann–Whitney $U$-test between each pair of treatments: *fas5* vs. GFP, *fas5* vs. WT, and GFP vs. WT. Significant effects in *fas5* knockdown (KD) females are indicated by up- (white) and downwards (black) arrows, corresponding to either up- or downregulation of the absolute compound quantities, respectively. Where compound identifications were ambiguous due to multiple possible methyl-

branch positions based on the detected ion pairs, all possible compound configurations are given.

• MDAR checklist

## Data availability

The datasets generated or analyzed during this study are available at the figshare data repository under https://doi.org/10.6084/m9.figshare.20411958.

The following dataset was generated:

| Author(s) | Year | Dataset title | Dataset URL | Database and Identifier |
|---|---|---|---|---|
| Sun W, Lange MI, Gadau J, Buellesbach J | 2023 | Decoding the genetic and chemical basis of sexual attractiveness in parasitoid wasps | https://doi.org/10.6084/m9.figshare.20411958 | figshare, 10.6084/m9.figshare.20411958 |

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
