## [Editor Report]

This important study reveals the genetic regulation of changes in cuticular hydrocarbon profiles in a Hymenopteran insect and links these changes with courtship behaviour and sexual attractiveness. It provides convincing empirical evidence, spanning genetic, chemical, and behavioural data. It adds valuable new perspectives on the mechanisms that underlie chemical recognition and communication systems in nature.

---

## [Decision Letter]

**Decision letter after peer review:**

Thank you for submitting your article "Decoding the genetic and chemical basis of sexual attractiveness in parasitic wasps" for consideration by *eLife*. Your article has been reviewed by 3 peer reviewers, and the evaluation has been overseen by a Reviewing Editor and Claude Desplan as the Senior Editor.

Essential revisions:

1) Improved clarity of statistical tests, in line with comments made by Reviewers.

2) A revised discussion of the results in males, in line with comments made by Reviewers.

You are encouraged to also consider the various other comments by reviewers when making your revision.

*Reviewer #1 (Recommendations for the authors):*

L127: Make it clearer that the 13.44% refers to Methyl position 3 and 47.5% to position 5 also indicate if you state the median or mean, adding standard deviation would further help to add context.

Figure 2B (and 4C and S2A). The changes in proportion are hard to compare between treatments from a stacked barplot. Consider showing the different compound classes rather in a side-by-side boxplot for each compound class across the 3 treatments.

Line 140: I don't agree that the pattern is similar in males. Unlike females, in Fas5 males there is also an increase in alkanes and a lack of change in 3Me and 9+Me at least between the two mutants. This has to be reworded and addressed accordingly. The lack of such results also contradicts to some extent your main conclusions in Line 200. If Fas5 regulates the up and down regulation of methyl groups at positions 3 and 5 differently than position 7+ why do you not see this pattern in males? The same problem is repeated in Line 268, you do not show the same effect for both sexes.

Line 226: Do you not mean "alkene fraction" instead of "MB fraction"?

L127: In figures 2C-F you show absolute amounts but in the main text you refer to relative abundances, why not use only one form of measurement for consistency?

L127: You refer to GFP RNAi females and later on you refer to GFP controls, if this is the same try to be consistent with your language.

Figure S1B does not seem to be mentioned in the main text, consider removing it.

The data availability for the raw data in Line 457 does not seem to work.

Stats:

Overall the Manuscript lacks any kind of detailed explanation of the statistical methods used nor are there any detailed statistical results mentioned throughout the manuscript. This makes it impossible to check if the significant differences mentioned in the manuscript are genuinely present, if the multiple comparisons are applied in a meaningful way, or if the used tests are appropriate for the distribution of the data (is the data parametric? Did they test for homogeneity of variance?). How was a Mann Whitney U test conducted on a GC figure (Figure 2A)? I strongly suggest the authors revise the statistics section with a much more thorough reporting of the tests used and their results before being able to properly evaluate the reported results.

For instance (but not limited to):

In Figure 2A: why did you not conduct an ANOVA to first test against general differences across groups and follow it up by a MANOVA (or similar) to test for significant differences across groups? A Random Forest analysis could also have elucidated the major compounds driving the differences across groups in a more statistically robust manner. I am also surprised that a standard method of illustrating differences across CHC groups (a PCA or NMDS) was not used in this study.

Line 644: why was a Mann Whitney U test directly performed (a posthoc test) and not a Kruskal Wallis test, or similar, for a first comparison across groups?

Table S2: While Table S2 states a detailed list of all the compounds it does not list any information on the stats.

Figure S3 How were the survival probabilities calculated? Considering the very low amount of data per treatment (N=10) maybe showing a true mortality curve with the real data would be more informative rather than trying to plot a model over it.

*Reviewer #2 (Recommendations for the authors):*

Knockdown methods: this sounds like the amount of solution that was injected was not quantified and only standardized visually using the food dye? Please specify.

I did not understand how the apparatus for the dummy experiments worked.

Figure 2: I found it confusing that the order of compound types in the chart is the opposite of that in the legend (in B).

Survival analysis: relatively low sample size although I agree there is no indication that the knockdown affects survival under any condition.

Table S2: indicate what statistics were used for testing significant changes in the individual substances. Correction for multiple testing?

There are a couple of small typos throughout the manuscript.

*Reviewer #3 (Recommendations for the authors):*

I would suggest the authors display current Figure 2 C to G as a supplementary figure and rather display the equivalent figures displaying the ratio of the compounds in the total blends (as in Figure 2.B, but showing individual variability). More generally I would encourage the authors to show data points (i.e. using partly transparent dots) for all of their figures for which the number of replicates is not too high. See: Weissgerber, T. L. et al. 2019 « Reveal, Don't Conceal » Circulation.

The authors provide convincing evidence that CHCs modification induced by their RNAi experiment does not alter the resistance of females to desiccation. However, they do not present the results in the methods section and refer to the data supporting this conclusion in the discussion and methods sections only. I believe the authors should briefly present the results regarding the desiccation experiments in the Results section.

Line 248 to 252, the authors compare the sexual attractiveness they observed with the one conveyed through female sex pheromones in moths. While their sentence is not wrong, I think it conveys a wrong message. In moths, female sex pheromones attract males usually through one or a few necessary and sufficient compounds. This attraction is further modulated (positively or negatively) by other compounds in the blends and their ratio. I would suggest the authors convey more clearly this in this part of the discussion.

---

## [Author Response]

Essential revisions:Reviewer #1 (Recommendations for the authors):L127: Make it clearer that the 13.44% refers to Methyl position 3 and 47.5% to position 5 also indicate if you state the median or mean, adding standard deviation would further help to add context.

We now include standard deviations, and also rephrased the sentence to make it unambiguously clear. Moreover, we now indicate the absolute amounts instead of proportions (see reply to comment further below). As for means/medians, we now also include additional supplementary boxplots indicating proportions for all relevant CHC compound classes to allow for a comprehensive assessment of both the relative and the absolute variations, which is also in line with the following comment of the reviewer (line 145-162).

Figure 2B (and 4C and S2A). The changes in proportion are hard to compare between treatments from a stacked barplot. Consider showing the different compound classes rather in a side-by-side boxplot for each compound class across the 3 treatments.

Thank you for the suggestion. We plotted the proportions of different compound classes separately in different sub-plots, as shown in Author response image 1 and 2. Mann-Whitney U-tests were used between each two of the three treatments (namely, fas5 vs GFP, fas5 vs WT and GFP vs WT). Afterwards, Benjamin-Hochberg procedure was performed to correct the resulting p-values from the previous tests. Author response image 1 was generated from female CHC data, while Author response image 2 was generated from male CHC data.

**Author response image 1. sa2fig1:** 

However, we still prefer to keep the stacked barplot (Figure 2B) accompanied by the violin plots of the absolute CHC compound class quantities, for the following reasons:We intend to show a quick to grasp overview of the shift in proportions, most prominently displayed for the methyl-branched alkanes, which the stacked barplot immediately communicates visually, and also corresponds very well with the chromatograms we depict in Figure 2A.

2. As we concordantly display the absolute proportions of the methyl-branched alkane compound classes in Figure 2 C-F, we already expand upon the proportional stacked barplot in the same figure which is intended to give the reader a more detailed insight into the compound classes shift, and we still think that this also conveys our experimental journey of discovery quite accurately.

We think that including the boxplots depicted as supplementary data would be redundant, as the dataset is already conveyed through the stacked relative barplots and the violoin plots of the absolute quantities.

Line 140: I don't agree that the pattern is similar in males. Unlike females, in Fas5 males there is also an increase in alkanes and a lack of change in 3Me and 9+Me at least between the two mutants. This has to be reworded and addressed accordingly. The lack of such results also contradicts to some extent your main conclusions in Line 200. If Fas5 regulates the up and down regulation of methyl groups at positions 3 and 5 differently than position 7+ why do you not see this pattern in males? The same problem is repeated in Line 268, you do not show the same effect for both sexes.

We now include a section where we greatly elaborate on the CHC patterns shifts in *fas5* RNAi knockdown males (line 167-178).

Line 226: Do you not mean "alkene fraction" instead of "MB fraction"?

We are sorry for the confusion. We meant the “proportion of alkene in the extracts of separated MB fraction”. We rephrased the sentence to make it more clear (line 297-298).

L127: In figures 2C-F you show absolute amounts but in the main text you refer to relative abundances, why not use only one form of measurement for consistency?

See above, we now include absolute amounts in the text as indicated in Figure 2C-G.

L127: You refer to GFP RNAi females and later on you refer to GFP controls, if this is the same try to be consistent with your language.

We changed all mentions of “GFP control” into “GFP dsRNAi females”.

Figure S1B does not seem to be mentioned in the main text, consider removing it.

We added the description of Figure S1B (now Figure 2—figure supplement 3B) in the result part (line 157).

The data availability for the raw data in Line 457 does not seem to work.

The link is now available.

Stats:Overall the Manuscript lacks any kind of detailed explanation of the statistical methods used nor are there any detailed statistical results mentioned throughout the manuscript. This makes it impossible to check if the significant differences mentioned in the manuscript are genuinely present, if the multiple comparisons are applied in a meaningful way, or if the used tests are appropriate for the distribution of the data (is the data parametric? Did they test for homogeneity of variance?). How was a Mann Whitney U test conducted on a GC figure (Figure 2A)? I strongly suggest the authors revise the statistics section with a much more thorough reporting of the tests used and their results before being able to properly evaluate the reported results.

We now include respective detailed descriptions of the statistical analyses we performed at the end of each sub-paragraph in the Materials and methods section.

For instance (but not limited to):In Figure 2A: why did you not conduct an ANOVA to first test against general differences across groups and follow it up by a MANOVA (or similar) to test for significant differences across groups? A Random Forest analysis could also have elucidated the major compounds driving the differences across groups in a more statistically robust manner. I am also surprised that a standard method of illustrating differences across CHC groups (a PCA or NMDS) was not used in this study.

We first performed a Shapiro test for normal distribution, which indicated that the data is not normally distributed (W = 0.592, p-value < 0.0001) and therefore a (strictly parametric) ANOVA would not have been permissible. Moreover, a Random Forest analysis or any form of “data distribution analysis” like NMDS, LDA and PCA is generally performed when the main differences in the variables are not immediately apparent or unclear. In our case, we directly observed a very conspicuous pattern change in chromatograms of *fas5* knockdown individuals and were thus able to access and confirm the individual variables significantly different between knockdowns and controls directly, which is actually quite rare in chemical ecological datasets. An NMDS reducing the multi-dimensional variable differences of the dataset to two dimensions simply depicted a shift in both *fas5* females and males as opposed to the control groups (see Author response image 3), but we have foregone depicting this result as we instead chose to immediately go into a more in-depth analysis about which variables/CHC compound classes are mainly shifted here, therefore we also first depicted the stacked barplot (see reply to reviewer comment above).

**Author response image 3. sa2fig3:** 

Line 644: why was a Mann Whitney U test directly performed (a posthoc test) and not a Kruskal Wallis test, or similar, for a first comparison across groups?

Actually, a Mann-Whitney U test can be used to directly compare the means of two datasets, while the Kruskal-Wallis test is traditionally applied when there are more than two means to compare. Although it can be argued that this is the case for our dataset at first glance, we reverted to a sequential Mann-Whitney U test with a Bejamini-Hochberg correction for multiple testing as this is generally considered the most powerful adjustment method in multiple testings (Benjamini and Yekutieli 2001) to give the most accurate, validated indication of the variables (*i.e.,* CHC compounds) significantly shifted between the knockdowns and controls.

Table S2: While Table S2 states a detailed list of all the compounds it does not list any information on the stats.

We have addressed the reviewer's comment by incorporating the statistical method used for comparing single compounds among different treatments in both the corresponding Materials and methods section and the table caption (line 489-502; 1012-1015).

Figure S3 How were the survival probabilities calculated? Considering the very low amount of data per treatment (N=10) maybe showing a true mortality curve with the real data would be more informative rather than trying to plot a model over it.

While we acknowledge the reviewer's concern regarding the sample size in the desiccation assay, we would like to clarify the experimental design. Each testing vial contained 10 individuals, allowing us to generate a robust survival curve. We had 10 replicates (vials) for each treatment, resulting in a total of 10 survival curves per treatment based on 100 individuals in total. We modeled the survival probability based on this adequately sized dataset, which is comparable to other desiccation resistance studies (Engl et al. 2018; Ferveur et al. 2018; Wang et al. 2022b).

We apologize for neglecting a detailed explanation of the survival analysis, which we now add to the statistical test section. The survival probabilities were calculated through the following steps: 1. the survival times were sorted in an ascending order; 2. the number of individuals “at risk” (in our study, death) at each time point was calculated; 3. the probability of survival at each time point was computed. Initially, the survival probability was set to 1. Afterwards, for each time point, the survival probability was adjusted based on the proportion of individuals who survive up to that time point; 4. multiply the survival probabilities across all time points to obtain the overall survival probability for each individual.

Reviewer #2 (Recommendations for the authors):Knockdown methods: this sounds like the amount of solution that was injected was not quantified and only standardized visually using the food dye? Please specify.

You are correct in pointing out that the exact volume of the double-stranded RNA (dsRNA) solution used for injections was challenging to quantify accurately. This was primarily due to the practical limitations encountered during the injection process that the opening of the needles was not fixed and became gradually blocked by the pupal tissue with each subsequent injection. To overcome this challenge, we employed the spread of a red dye within the pupal abdomen as a visual indicator, as demonstrated in Author response image 4. The procedure is also described in Wang et al. (2022a), and we now refer to this publication at the corresponding part in Materials and methods.

**Author response image 4. sa2fig4:** 

I did not understand how the apparatus for the dummy experiments worked.

We apologize for the confusion. Our behavior assay setup is illustrated in Author response image 5 (left) and derived from Buellesbach et al. (2014) as indicated. The setup consists of two aluminum plates, each containing 12 mating chambers. In the experimental procedure, we initially placed single freeze-killed females into each of the 12 chambers in one plate. Subsequently, single WT males were introduced into each hole in the opposite plate and promptly covered with glass slides. Following this, the behavioral assays were initiated by swiftly adjoining the two plates. The recording was then immediately initiated. Author response image 5 on the right side provides an overhead view of the setup.

**Author response image 5. sa2fig5:** 

Figure 2: I found it confusing that the order of compound types in the chart is the opposite of that in the legend (in B).

We changed the order of the compound types in the figure.

Survival analysis: relatively low sample size although I agree there is no indication that the knockdown affects survival under any condition.

We acknowledge and agree with your comment. For detailed clarification of the experimental design and survival analysis conducted, we kindly refer you to our response provided to reviewer 1.

Table S2: indicate what statistics were used for testing significant changes in the individual substances. Correction for multiple testing?

We incorporated the statistical method used for comparing single compounds among different treatments in both the corresponding Materials and methods section and the table caption (line 489-502; 1012-1015).

There are a couple of small typos throughout the manuscript.

We went through the manuscript and corrected all remaining typos.

Reviewer #3 (Recommendations for the authors):I would suggest the authors display current Figure 2 C to G as a supplementary figure and rather display the equivalent figures displaying the ratio of the compounds in the total blends (as in Figure 2.B, but showing individual variability). More generally I would encourage the authors to show data points (i.e. using partly transparent dots) for all of their figures for which the number of replicates is not too high. See: Weissgerber, T. L. et al. 2019 « Reveal, Don't Conceal » Circulation.

We kindly refer you to response to reviewer 1, where we show the plots of compound ratios as suggested, but we would still prefer to keep Figure 2 C-G as we strongly think that this communicates our main findings and message best.

The authors provide convincing evidence that CHCs modification induced by their RNAi experiment does not alter the resistance of females to desiccation. However, they do not present the results in the methods section and refer to the data supporting this conclusion in the discussion and methods sections only. I believe the authors should briefly present the results regarding the desiccation experiments in the Results section.

We appreciate this suggestion, and now include the figure as well as the outcome description of our desiccation assays in the Results section (line 227-234).

Line 248 to 252, the authors compare the sexual attractiveness they observed with the one conveyed through female sex pheromones in moths. While their sentence is not wrong, I think it conveys a wrong message. In moths, female sex pheromones attract males usually through one or a few necessary and sufficient compounds. This attraction is further modulated (positively or negatively) by other compounds in the blends and their ratio. I would suggest the authors convey more clearly this in this part of the discussion.

We agree with the reviewer’s valid point, and now re-wrote that part to focus on CHC variation and its impact on sexual attractiveness within the scope of the evolution of Hymenoptera, to not extrapolate this quite specific communication modality too much and remove speculations on similarities to the differentially functioning moth pheromonal communication (line 321-327).